# Impact on Mechanical Properties and Microstructural Response of Nickel-Based Superalloy GH4169 Subjected to Warm Laser Shock Peening

**DOI:** 10.3390/ma13225172

**Published:** 2020-11-16

**Authors:** Ying Lu, Yuling Yang, Jibin Zhao, Yuqi Yang, Hongchao Qiao, Xianliang Hu, Jiajun Wu, Boyu Sun

**Affiliations:** 1Shenyang Institute of Automation, Chinese Academy of Science, Shenyang 110016, China; luying@sia.cn (Y.L.); yangyuqi@sia.cn (Y.Y.); hcqiao@sia.cn (H.Q.); huxianliang@sia.cn (X.H.); wujiajun@sia.cn (J.W.); sunboyu@sia.cn (B.S.); 2Institutes for Robotics and Intelligent Manufacturing, Chinese Academy of Sciences, Shenyang 110169, China; 3College of Science, Northeastern University, Shenyang 110819, China

**Keywords:** warm laser shock peening (WLSP), GH4169 nickel-based superalloy, microstructure, residual stress

## Abstract

Laser shock peening (LSP), as an innovative surface treatment technology, can effectively improve fatigue life, surface hardness, corrosion resistance, and residual compressive stress. Compared with laser shock peening, warm laser shock peening (WLSP) is a newer surface treatment technology used to improve materials’ surface performances, which takes advantage of thermal mechanical effects on stress strengthening and microstructure strengthening, resulting in a more stable distribution of residual compressive stress under the heating and cyclic loading process. In this paper, the microstructure of the GH4169 nickel superalloy processed by WLSP technology with different laser parameters was investigated. The proliferation and tangling of dislocations in GH4169 were observed, and the dislocation density increased after WLSP treatment. The influences of different treatments by LSP and WLSP on the microhardness distribution of the surface and along the cross-sectional depth were investigated. The microstructure evolution of the GH4169 alloy being shocked with WLSP was studied by TEM. The effect of temperature on the stability of the high-temperature microstructure and properties of the GH4169 alloy shocked by WLSP was investigated.

## 1. Introduction

The nickel-based superalloy has been widely used as turbine blade and disk link material in aircraft, mainly due to its excellent thermal mechanical properties stability, such as thermal fatigue, rupture ductility, oxidation resistance, and creep strength. However, with the development of new aircraft engines, higher requirements for mechanical properties have been put forward to improve the service stability and life of turbine blades or disks. 

Investigations have shown that temperature has an important influence on the flow stress of metal materials, mainly in the following aspects [1,2,3,4,5]:
Dislocation motion: Increase of temperature enhances thermal activation, and the concentration of vacancy defects is reduced in the crystalline grain. The edge dislocation begins climbing, which prompts the generation and annihilation of vacancy. The local equilibrium concentration of the dislocation is maintained, and the slippage movement of the screw dislocation also requires sufficient energy to activate.Critical shear stress: Higher temperature leads to critical shear stress, which results in material decrease and slip system increase. With the increase of temperature, atomic momentum becomes larger, and interatomic force is damaged, and then shear stress is reduced. There are more sliding positions, and the slip system is more active.Organization of metal materials: As temperature increases, the solubility of solute atoms increases, which results in the reduction of the resistance of the alloy. The material thermoplastic and atomic thermal vibration are reinforced, and atoms in the lattice become unstable. In this state, the atoms are easy to move along the gradient of the stress field under the action of external forces, and the plasticity becomes more strenuous.

It is clear that excellent mechanical property of the material can be achieved by increasing the temperature during treatments.

Warm laser shock peening (WLSP) combines the advantages of both laser shock peening (LSP) and dynamic strain aging (DSA); hence, WLSP technology can improve the fatigue properties of materials under alternating loads and high temperatures. WLSP is beneficial to the stabilization of the dislocation structure of materials and suppression of the release of residual stress induced by LSP. Compared with the traditional LSP technology, WLSP is carried out at high temperature rather than room temperature. This in turn can achieve multifaceted mechanical performance optimization [6,7,8]. By increasing the temperature, dynamic strain aging occurs, a deeper residual stress layer on the surface of the material is formed, and nanoprecipitates are generated, which results in a significant increase in the pinning effect on dislocation and stability. Meanwhile, the density of the dislocation structure can be improved, and a more stable residual compressive stress can be achieved by WLSP. This effectively suppresses the effect of high-temperature instability and improves the high-temperature stability of the material surface’s residual compressive stress layer, which is beneficial to the improvement of fatigue life. Hence, WLSP has been looked into as a new and effective technology that can be used to improve the mechanical properties of materials [9,10,11]. So far, most of the research studies of the WLSP strengthening technology have mainly focused on low-temperature alloy materials. It is well known that high-temperature alloys based on iron, nickel, and cobalt are key materials used for hot end parts in the fields of aerospace, aviation, chemical industry, and energy industry [12,13,14]. Hence, investigations on improving the performances of such materials processed by WLSP are of major and immediate practical significance. The application of the GH4169 superalloy was restricted under the temperature of 650 °C due to the coarsening and transformation of the γ” phase processed under a higher temperature. Since WLSP is carried out under 650 °C, the coarsening and transformation of the γ” phase would be inhibited. Therefore, WLSP is a preferred technology used to improve the properties of GH4169. To our knowledge, few efforts have been put on investigations on improving the surface structure stability and high-temperature service performance of the GH4169 superalloy by the WLSP technique [15,16,17].

In the present work, LSP processing under 25 °C and WLSP processing under 150, 200, and 250 °C were carried out. The surface morphology, stress, and microstructure of the GH4169 alloy before and after LSP and WLSP processing were studied using theoretical and experimental methods. The effect of temperature on the stability of the high-temperature microstructure and properties of the GH4169 alloy processed by WLSP was investigated.

## 2. Experiment Details

### 2.1. Materials and Microstructure Observation

The nickel-based GH4169 superalloy was used as experimental metallic material. Its chemical composition (wt %) is 51.58 Ni, 18.36 Cr, 20.17 Fe, 2.96 Mo, 5.03 Nb, 0.99 Ti, and 0.18 Si (which is showed in Table 1.). In the alloy, Ni is the main element that forms the austenite matrix, and other elements, such as Fe and Co dissolved in the matrix, play a role in solid solution strengthening. The compounds formed with Nb and Ti can be precipitated as the strengthening phase in the matrix. Meanwhile, the transition of the metastable γ” phase to the equilibrium δ phase could be inhibited by the addition of Nb and Ti. Furthermore, the addition of Al, Ti, Cr, and Mn has the function of improving the corrosion resistance and oxidation resistance of alloys. The specimen was cut into a rectangular shape with a size of 20 × 20 × 10 mm^3^. The surfaces were mechanically polished to eliminate residual tensile stress prior to LSP treatment, and then peened by LSP roughly parallel to (001) orientation.

### 2.2. Warm Laser Shock Peening Process

The laser shock temperature was decided based on the formula of “qian–xiaoli model” proposed by Qian [18]. According to his theory, the temperature range of the dynamic strain aging of alloy is approximately 0.2–0.5 Tm (Tm is the melting point temperature of an alloy). In this work, the melting temperature of the GH4169 alloy is 1260 °C; In contrast, the normal temperature of LSP was set to 25 °C.

A layer of aluminum foil sprayed with K9 glass with a thickness of about 3 mm was pasted on the target surface as an ablative layer. The glass layer prevented the shock wave from spreading and prolonged the effect of the shock wave. The samples were put on the pedestal, fixed by customized fixtures, and then heated to the setting temperature. The laser and robot were opened simultaneously as soon as the temperature reached the setting value. The processing was performed according to the default trajectory as illustrated in Figure 1. The samples were removed after being cooled down to room temperature, and the surface was cleaned with alcohol in an ultrasonic cleaner for 40 min.

The Nd:YAG laser system (Beamtech, Beijing, China) with a wavelength of 1064 nm and an average power of 130 W was used as the laser shock peening processing device. The laser parameters were set at a pulse width of 14 ns, pulse energy of 5 J, and repetition rate of 2 Hz. The spot diameter was focused at 2 mm, and the overlap rate was 50%. The heating system included a heating pedestal, fixture, temperature sensor, XMTD-2501 temperature controller (Tongyong, Shenyang, China), and relay. The realization of the machining trajectory was achieved by a KUKA KR 270 R2900 ultra K model 6 axis robot (KUKA, Augsburg, Germany), which ensured a precise motion of 0.1 mm in the Cartesian coordinate system. The snake-crawling trajectory used in the WLSP experiment is shown in Figure 1.

### 2.3. Measurements of Mechanical Properties and Microstructure

The surface residual stress was measured using a D/Max-2500PC X-ray diffraction equipment (Rigaku, Tokyo, Japan). The radiation source was CuKα, and the X-ray beam diameter was 1 mm.

The microhardness of the shocked surface and cross-sectional depth was tested using a Micro Vickers Hardness Tester (DHV-1000, Time, Beijing, China) with a 0.98 N load and 10 s dwelling time. The microhardness along the cross-sectional depth direction was measured at several positions with a distance of 10 µm interval. At each location, three different points were selected to get the average value of the microhardness.

The evolution of the microstructure of the samples was investigated by TEM (JEM-2100, JEOL, Rokumaru, Japan) operated at 200 kV. The TEM samples were prepared by grinding the substrate to a thickness less than 50 µm and polishing with a double-jet polisher at −35 °C in a solution consisting of 10% perchlorate acid + 25% ethanol + 65% n-butanol.

The fatigue specimen was cut and cleaned ultrasonically for 20 min, and the fracture morphology was observed using SEM (JSM-6010LA, JEOL, Rokumaru, Japan).

## 3. Results and Discussion

### 3.1. Surface Morphology

The surface morphology was observed with a light interference instrument and is illustrated in Figure 2. As shown in Figure 2a, a uniform distribution of loose tissue defect and hard point bumps on the surface of GH4169 without peening was observed. The hard point bumps were formed by the strengthening carbide phase (γ, γ′′, γ′, δ), which are called the MC (MC, a kind of carbide precipitates) organization precipitated out during solidification. After being polished, this phase maintained the shape of the point bumps. Investigations showed that such characteristic of the tissue would contribute to high hardness and strong deformation capability.

In Figure 2, the Figure 2b,c illustrates the surface morphology of the surfaces after LSP and WLSP, respectively. Compared with the surface of the sample processed with LSP (Figure 2b), it is clear that no obvious relief structure was observed on the surface of the sample processed with WLSP as shown in Figure 2c. Instead, a lot of surface ripples appeared. Meanwhile, the ripples became more homogeneous with increasing peening times and laser energy, as illustrated in Figure 3. The surface roughness was quantitatively analyzed. According to the results of the ripples on the surface of the sample processed with WLSP, as shown in Figure 3b (peened four times), the roughness Rz was 9.449 µm, and the depth reached 1.5 µm, which could be easily distinguished. Compared with the surface reliefs peened at room temperature (Rz 1.8 µm), the roughness of the samples processed with WLSP was much higher.

Surface reliefs by laser shock is a common phenomenon at room temperature. There is no clear explanation for this phenomenon so far. Based on our research, we concluded following possible explanations: During laser shock peening treatment, a plastic deformation delay occurred. The plastic deformation depends on the energy distribution of the laser. In this work, the temperature distribution of the laser beam could be characterized as Gaussian distribution. The uneven distribution of laser spot energy may result in uneven plastic deformation delay in different locations. This in turn may lead to relief formation on the surface. Furthermore, a standing wave was formed between the constrained layer and the sacrifice layer due to wave superposition during the LSP process. The uneven distribution of the temperature field and energy of the standing wave led to the formation of reliefs on the surfaces. Besides, the strengthening phase action may be another reason for the relief’s formation. In combination with the surface ripples formed by WLSP processing, the author prefers the theory that the strengthening phase plays a major role in the evolution of the surface topography of GH4169.

### 3.2. Microhardness Evaluation

As shown in Figure 4, the microhardness for the base material is about 374 HV0.1. The microhardness presents different values after WLSP at different temperatures with same laser energy (5 J) and the same peened times.

The microhardness distribution along the cross-sectional depth direction is shown in Figure 4. It was observed that the temperature had great influence on the hardness value. The surface microhardness increased to 440, 480, and 492 HV, improving by 17.6%, 28.3%, and 31.6% after being processed with LSP temperatures of 25 (room temperature, RT), 250, and 300 °C, respectively (three points were tested each time in the same parameters, and the standard deviation was small). It is obvious that the microhardness was much more significantly improved by WLSP than LSP at RT. The microhardness then gradually decreased along the cross-sectional depth to the value of the substrate region. The thickness of the hardened layer increased with the increase of the WLSP temperature. The hardened layer could reach 1.6 mm peened at 250 °C, compared with the samples peened at room temperature, which was only about 1 mm. The depth resulting from plastic deformation was proportional to the laser shock pressure. This relationship between microhardness and temperature was nonlinear.

The increase of the microhardness of the surface can be attributed to the formation of a typical surface structure after LSP and WLSP treatment. As stated earlier, a plastic deformation occurred during LSP or WLSP, which resulted in the increase of dislocation density and the refinement of the surface grains. Meanwhile, the plastic deformation, especially the high-speed plastic deformation, resulted in a strain. The dislocation density, the refinement of the the grains, and the stain due to the plastic deformation, in turn, enhanced the microhardness of the surface. However, the hardness of WLSP differed from that of LSP. Despite the plastic deformation and grain refinement similar to LSP, dynamic strain aging (DSA) is another important factor that improves the microhardness through WLSP treatment. It is well known that the highly dense γʹʹ phase would be generated during WLSP due to high-speed deformation and dynamic precipitation. The volume density of γʹʹ precipitates depends on the temperature and laser power. A higher temperature during WLSP can dramatically increase the volume density and provide higher thermal energy, which will accelerate the precipitation process. A higher temperature provides higher energy for the atoms to solute in the alloy.

With the increase of temperature, the energy of the solute atoms in the alloy increases. Dislocation pinning will occur when the activation energy of the atoms is high enough and the moving speed of the atoms is less than that of the dislocation. This, in turn, leads to the accumulation and proliferation of the dislocation to get a higher dislocation density and grain refinement. The higher temperature results in the softening of the material, as well as the decrease of the elastic modulus. Therefore, the microhardness of the surface increased significantly after peening. At the same time, the damping was reduced, resulting in the further propagation of the shock wave within the materials and the formation of a thicker surface strengthening layer.

### 3.3. Analysis of Residual Stress

For the purpose of evaluating the stability of the samples processed by LSP and WLSP, the residual stress at different temperatures was measured by the XRD method, and the results are illustrated in Figure 5 (the red line is the surface residual stress shocked by WLSP at 250 °C, and the black one is the surface residual stress shocked by WLSP). It was observed that the surface residual stress due to WLSP was slightly lower than that due to LSP at any aging temperature as shown in Figure 5a–c (surface residual stress via exposure time at different temperatures of (a) 600, (b) 650, and (c) 700 °C). Similar trends were found among the samples processed at different aging temperatures.
The surface residual stress shocked by WLSP was slightly lower than that shocked by LSP. However, the release of residual stress during the post-aging process induced by WLSP was less than that induced by LSP, indicating that the samples processed with WLSP were more stable than that processed with LSP.The relaxation effects of the residual stress on the surface increased gradually with the increase of the aging temperature. No significant change in the surface residual stress was observed during the aging process at a temperature of 600 °C. The residual stress induced by LSP released completely, while that induced by WLSP only decreased by 8% when the aging temperature increased to 650 °C. Moreover, when the aging temperature increased to 700 °C, the residual stress induced by LSP released soon, while that induced by WLSP only decreased by 50% after 100 h.

It was found that both the residual stress and the dislocation density increased due to the increase of temperature and the laser intensity. On the one hand, with the increase of the aging temperature, the motion resistance of dislocation decreased, the dislocation entanglement gradually expanded, and dislocation rearrangement and annihilation appeared, which resulted in a decrease of dislocation density and lattice distortion. With the increase of the temperature, the residual stress relaxed rapidly. On the other hand, the increase of the aging temperature resulted in the increase of the grain size of the γ” phase and the transformation of the γ” phase to the δ phase. The results are in accordance with the surface hardness test results. The results further confirmed that the enhanced pinning effect caused by a higher volume density of γ” precipitates was responsible for the temperature and laser intensity effects on the hardening ratio. As a result, the strength and resistance of the material decreased quickly with the wide relaxation of the residual stress during the aging procedure under high temperature. The magnitude and stability of laser-induced residual stress played critical roles on the determination of the fatigue performance of metallic materials. In summary, the higher the aging temperature, the better thermal stability of the stress/tissue produced by WLSP.

### 3.4. Microstructure Analysis

It is well known that the properties of a certain material depend on its microstructure. The nickel-based GH4169 superalloy is mainly composed of the γ phase (fcc, face centered cubic) as the matrix, auxiliary reinforcement phase γ’-Ni_3_Al (fcc, intermetallic compounds), reinforcement phase γ’’-Ni_3_Nb (bcc, body centered cubic), and δ phase (stable phase). There will be a transformation from the γ’’ phase to the stable δ phase during a long-time aging processing or serving. Accordingly, the strength will reduce sharply. The TEM observation results (Figure 6) illustrated that the internal part of GH4169 after stress relief annealing presented lower dislocation density and a small amount of lattice distortion distributed around the partially reinforced phase. A small amount of the disk-shaped γ” phase and the granular γ” phases distributed in the crystal was observed. Besides, a large amount of spherical or platelike δ phase precipitates was also observed on the grain boundaries. We deduced that such highly “tangled” structures would be beneficial for the stability of the mechanical properties at high temperature (about 650 °C). When the dislocation density reaches a certain threshold, dynamic recrystallization occurs at the grain boundaries, shear bands, and impurities, which promotes the dynamic precipitation or refinement of the strengthening phases.

The appearance of DSA and DP (dynamic precipitation) on the surface of the material during WLSP would have an effect on the microstructure and in turn improve its properties. By comparing the differences in Figure 7, Figure 8 and Figure 9, we can conclude the following:

(1) Different from the structures in Figure 6 corresponding to the sample before peening, there were a large number of complex dislocation structures in the γ phase matrix after laser shock peening (Figure 7a). After LSP, the dislocation organization obviously increased, the dislocation density greatly increased, and many substructures were formed. The red line illustrated in Figure 7a shows the direction of the grain deformation under laser shock, and the yellow lines represent the direction of the dislocations in the γ phase matrix. The angle between the red line and the yellow line is approximately 90°. The dislocations in the δ phase were produced in large quantities at the edge of the grain boundaries and moved towards the interior of the grains (Figure 8a). It was observed that dislocation twins appeared along the long axis direction of the δ phase. The twins formed along the same direction with the dislocation movement. A large number of dislocations accumulated at the grain boundaries, and the aggregation of dislocations at the grain boundaries might further cause grain boundary deflection (Figure 9a). A large number of dislocation lines appeared in the phase and moved along the grain boundaries. The accumulation of a large number of dislocations at the grain boundaries may further cause grain boundary deflection.

(2) The peening under high temperature in the process of WLSP resulted in the continuous accumulation of dislocations in the γ phase matrix and the formation of a large number of entangled dislocations (Figure 7b). The number of dislocations and their density in the samples processed by WLSP increased by a large margin compared with that in the sample processed by LSP. The effect of the dislocation on the grains or grain boundaries was more obvious, and a lot of dislocations gathered near the grain boundaries. Especially when the temperature reached 150 °C (Figure 9b), the dislocation accumulation caused a significant deflection of the grain boundaries. When the samples were shocked at a temperature of 200 °C (Figure 9c), some of the crystal boundaries darkled away so that it was difficult to identify them due to the action of the dislocations. The concentration of the dislocation structures could provide nucleation sites for phase precipitation, which was beneficial to dynamic precipitation (Figure 10b).

During laser shock peening, the dislocations moved towards a single direction and eventually gathered at the grain boundaries. Then they were blocked by the boundaries, which resulted in the formation of twins, subgrain boundaries, and new grain boundaries. Ultimately, a large number of crystal defects were formed. These defects moved and aggregated, and a part of them formed a subgrain boundary at a small angle. When they moved to the phase interface, they were pinned by the interface surface, and the migrations of the subgrain boundaries were obstructed, which could eventually lead these subgrain boundaries to gather at the phase boundary. The grain boundaries would be deflected under the interaction of the phase boundaries and the surface tension of the subgrain boundaries to balance the surface tension. (The aggregating dislocation morphology is as shown in Figure 9b.)

It is well known that the strengthening γ″ phase is a metastable phase. It will coarsen and transfer to a stable δ phase under a high temperature condition. The transformation then leads to a rapid change in strength and creep property. The γ” phase precipitated in the δ phase retarded the dislocation motion and the nucleation and growth of the recrystallized grains. Meanwhile, the accumulation of dislocations limited the growth of the δ phase. As stated earlier, the surface processed by WLSP presented superior high-temperature stability in residual stress than that processed by LSP. The reason is the DSA phenomenon caused by the increase of temperature. Although the resistance of the dislocation motion decreased at high temperature, the solute atoms diffused quickly enough to catch up with the motion of the dislocations to form a mass of solute atoms with large size enough to pin the dislocations. The reduction of the movable dislocations promoted the proliferation of a large number of dislocations, and finally increased the density of the dislocations. Meanwhile, the dislocation annihilation’s negative effect caused by dynamic recovery was the smallest, and the dislocation density also kept a larger value at high temperature, which improved the high-temperature stability of the material. The pinning effect of the secondary phases in the GH4169 alloy was analyzed, and it can be concluded that the precipitated γ” phase within the grains retained the dislocation motion, the precipitated γ” phase in the δ grains hindered the nucleation and growth of recrystallized grains, and the dislocations limited the δ grain growth.

Comparing the bright and dark fields of the microstructure features shocked by WLSP and LSP (Figure 11), a certain number of dislocations were accumulated in the γ′′ phase, and a large number of subphases appeared in the γ′′ phase, which finally led to the disk-shaped γ′′ phase being cut off. The γ” phase (the size was about 100 nm) was cut into a smaller size of γ’ phases (the size was about 30 nm). This phenomenon can be explained as follows.

First, due to the pinning effect of the precipitates on the dislocations and the effect of dislocation proliferation, high-strain-rate deformation produced higher dislocation density, when it shocked at 200 °C. At the same time, the surface plastic deformation of the material was strengthened, which promoted the appearance of super vacancy [16]. The diffusion of the vacancy atom in the thermal insulation process reduced the system energy, which resulted in the nonequilibrium segregation of Nb and the increase of the nucleation position of the γ” phase and the nucleation rate of the γ” phase in the strengthening layer. The plastic deformation caused by WLSP promoted the appearance of super vacancy. The vacancy formed in the WLSP process combined with the Nb atoms to form a vacancy–Nb pair. The vacancy–Nb pair diffused towards the defect and reduced the system energy, resulting in the Nb nonequilibrium. Because the driving force of this concentration gradient would lead to the movement of the γ”/δ interface, heat treatment at a lower temperature would promote the expansion and spheroidization of the decomposed α phase [18,19].

The plastic deformation at warm temperature caused the plastic deformation on the surface of the material, which could lead to the appearance of supervoids. In the process of WLSP, the vacancy combined with the -Nb atom to form the vacancy–Nb pair [20]. The concentration gradient forces caused the γ”/δ interface to move, and the decomposed δ phases expanded and spheroidized at lower temperatures (the relationship between the grain boundary phase and its adjacent phase satisfied the orientation relationship of Burgers, as showed in Figure 12) [21].

On the other hand, the decomposition of the γ″ phase effect introduced by WLSP promoted the precipitation of the high density of nanoscale particles. The tangled dislocation band and the strength phase effectively inhibited the movement of the dislocations through the pinning effect, which contributed to the enhancement of the material stability at high temperature. After the decomposition, the dislocations climbed and slipped inside the matrix phase [12]. Uniform dislocations were rearranged as smaller-angle grain boundaries or subgrain boundaries perpendicular to the sliding surface, and the inhomogeneous dislocations were counteracted by each other, eventually increasing the size of the subgrain boundary area, which could lead to decreased grain size and increased surface hardness and yield strength [16]. (The influence of strength grain size on the yield flow stress and hardness is usually written in the form of the Hall–Petch relationship). The precipitated relative dislocation played a pinning role, and the accumulation of dislocations provided more nucleation sites for the precipitated phase. The precipitation and the abundant dislocation would inhibit the moving dislocations’ slip. The pinning effect promotes the microstructure to be more stable. Besides, the precipitations accelerate the grain refinement. WLSP decreased the phase content and the pinning force to the grain boundary migration. The dislocations in the elongated grains were clustered near the phases. The small-angle grain boundaries in the elongated grains formed new large-angle grain boundaries through migration and rotation, and the original elongated grains were divided into several smaller equiaxed grains [14,16,17,18]. Thus, the finely dispersed and dispersed γ″ phase was interspersed with a large number of dislocations in the δ phase, forming a structure similar to reinforced concrete. This structure could be very stable at high temperatures in the high-temperature stress release process, the dislocations had an excellent pinning effect and effectively improved the softening resistance and creep resistance at high temperature [22].

## 4. Conclusions

In this paper, the microstructure of the GH4169 nickel superalloy processed by WLSP technology with different laser parameters was investigated. The proliferation and tangling of dislocations in GH4169 were observed, and the dislocation density increased after WLSP treatment. The influences of different treatments by LSP and WLSP on the microhardness distribution of the surface and along the cross-sectional depth were investigated. The microstructure evolution of the GH4169 alloy being shocked with WLSP was studied by TEM. The effect of temperature on the stability of the high-temperature microstructure and properties of the GH4169 alloy treated with WLSP was investigated, and the analysis is as follows: (1)Compared with LSP, WLSP could obtain a more complex dislocation structure. The dislocation promoted the precipitation of the enhanced phase, and this, in turn, enhanced the pinning effect of the dislocation. The effect of the dislocation on the grains or grain boundaries was more obvious, and a lot of dislocations gathered near the grain boundaries, especially when the temperature reached 150 °C, and the dislocation accumulation caused a significant deflection of the grain boundaries. When the samples were shocked at 200 °C, with the action of the dislocations, some of the crystal boundaries were blurred, and it was difficult to identify them. A large concentration of dislocation structures could provide nucleation sites for phase precipitation, which is beneficial to dynamic precipitation (DP).(2)On the one hand, the surface plastic deformation of the material was strengthened, which promoted the appearance of super vacancy. The diffusion of the vacancy atom in the thermal insulation process reduced the system energy, which caused the nonequilibrium segregation of -Nb and increased the nucleation position of the γ” phase. Therefore, the nucleation rate of the γ” phase in the strengthening layer was increased. On the other hand, the uniform dislocations were rearranged as smaller-angle grain boundaries or subgrain boundaries perpendicular to the sliding surface, and the inhomogeneous dislocations were counteracted by each other, eventually increasing the size of the subgrain boundary area, which could lead to decreased grain size and increased surface hardness and yield strength.(3)The precipitated relative dislocation played a pinning role, and the accumulation of dislocations provided more nucleation sites for the precipitated phase. Thus, the finely dispersed and dispersed γ″ phase was interspersed with a large number of dislocations in the δ phase, forming a structure similar to reinforced concrete. This structure could be very stable at high temperatures, and with the high-temperature stress release process, the dislocations had an excellent pinning effect and effectively improved the softening resistance and creep resistance at high temperature.

## Figures and Tables

**Figure 1 materials-13-05172-f001:**
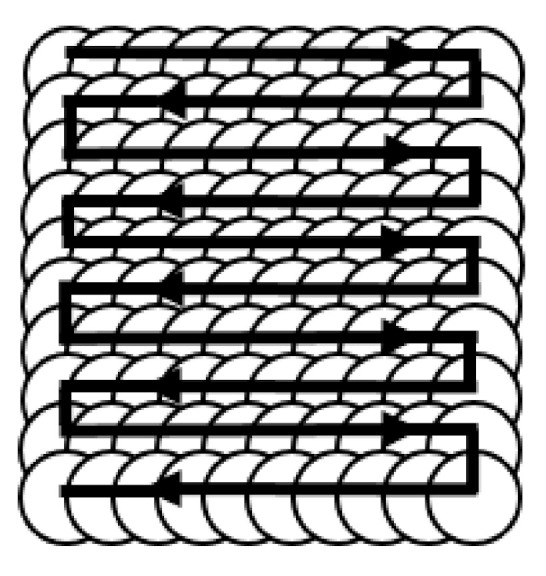
Snake-crawling trajectory used in the WLSP experiment.

**Figure 2 materials-13-05172-f002:**
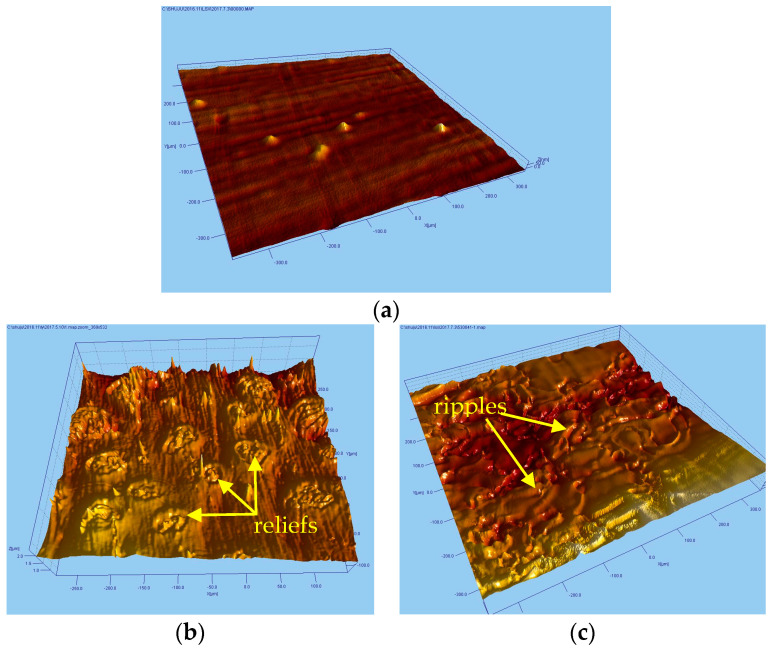
The surface morphology of different surfaces after the treatment of (**a**) the original surface, (**b**) LSP, and (**c**) WLSP.

**Figure 3 materials-13-05172-f003:**
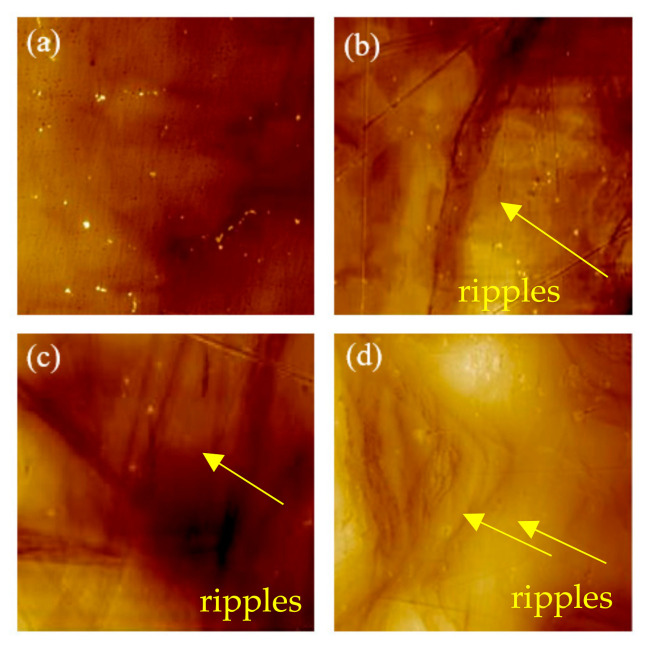
Local special surface morphology of specimen after WLSP: (**a**) 250 °C, 3 J, 1 time, spot overlap rate of 50%; (**b**) 250 °C, 3 J, 4 times, spot overlap rate of 50%; (**c**) 250 °C, 5 J, 1 time, spot overlap rate of 50%; (**d**) 250 °C, 5 J, 4 times, spot overlap rate of 50%.

**Figure 4 materials-13-05172-f004:**
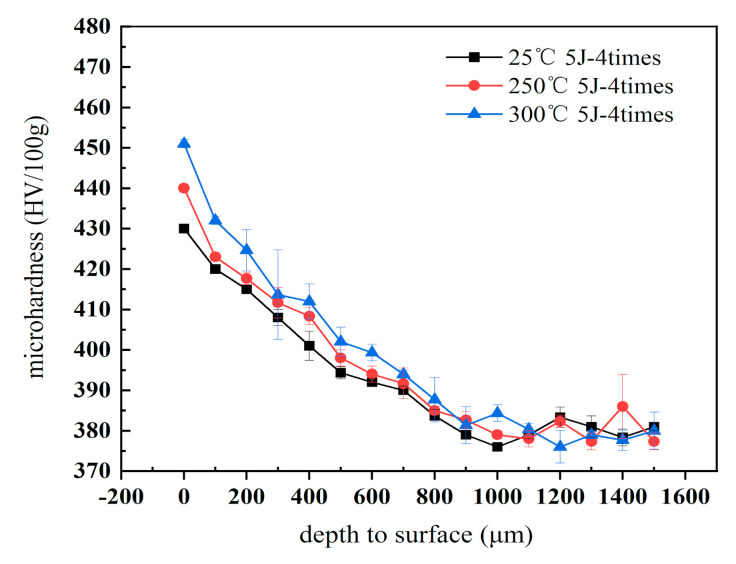
Microhardness distribution along the cross-sectional depth direction.

**Figure 5 materials-13-05172-f005:**
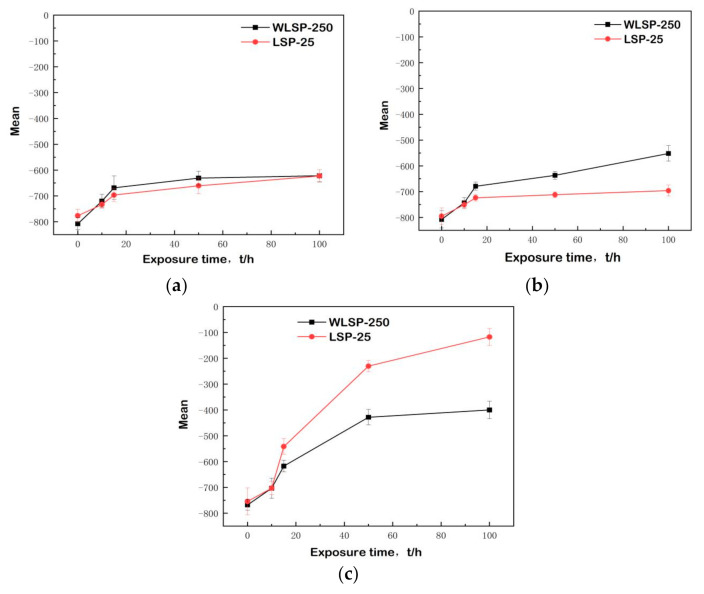
Surface residual stress via exposure time at different temperatures: (**a**) 250 °C, 3 J, 4 times, spot overlap rate of 50%, aging process at 600 °C; (**b**) 250 °C, 3 J, 4 times, spot overlap rate of 50%, aging process at 650 °C; (**c**) 250 °C, 3 J, 4 times, spot overlap rate of 50%, aging process at 700 °C.

**Figure 6 materials-13-05172-f006:**
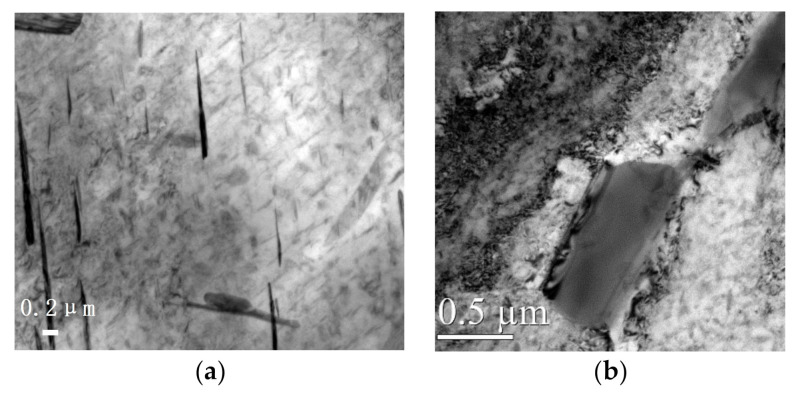
The main phase composition of GH4169. (**a**) γ-phase matrix, (**b**) δ phase.

**Figure 7 materials-13-05172-f007:**
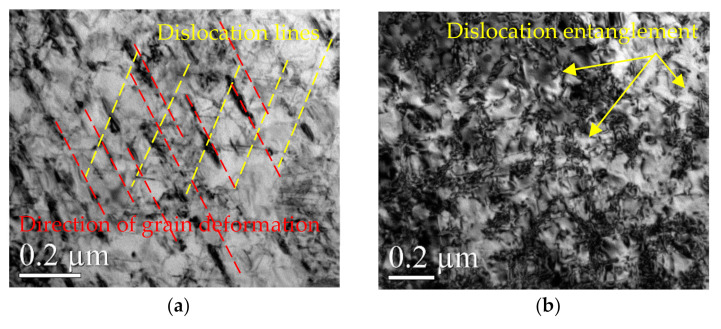
The change of the γ phase shocked by LSP at different temperatures: (**a**) 25 °C, (**b**) 200 °C.

**Figure 8 materials-13-05172-f008:**
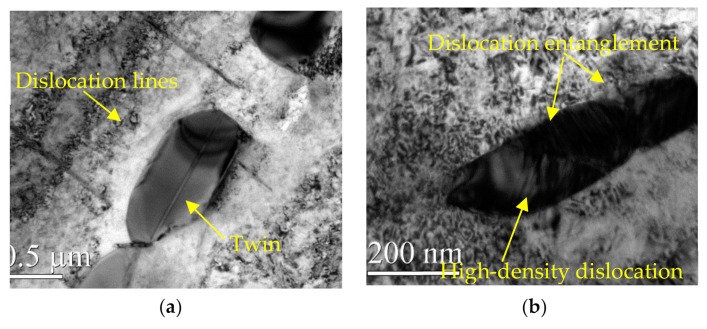
The change of the δ phase shocked by LSP at different temperatures: (**a**) 25 °C, (**b**) 200 °C.

**Figure 9 materials-13-05172-f009:**
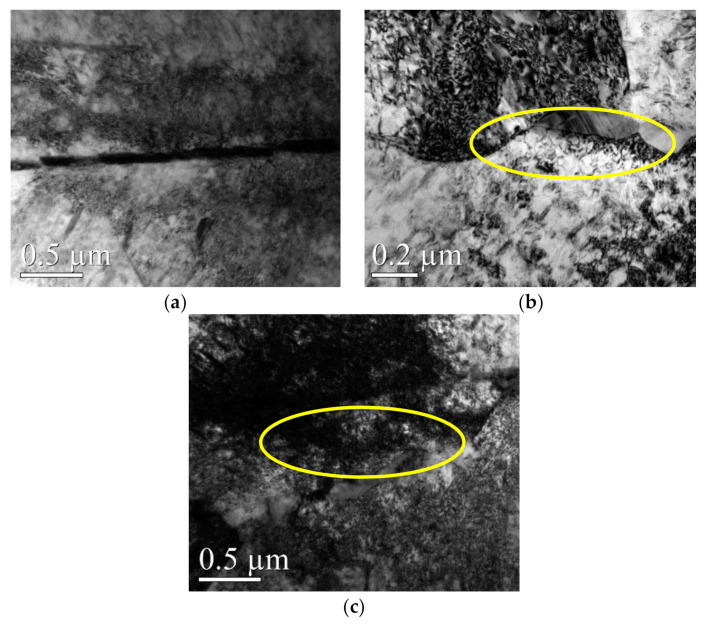
The change of the grain boundaries shocked by LSP at different temperatures: (**a**) 25 °C, (**b**) 150 °C, (**c**) 200 °C.

**Figure 10 materials-13-05172-f010:**
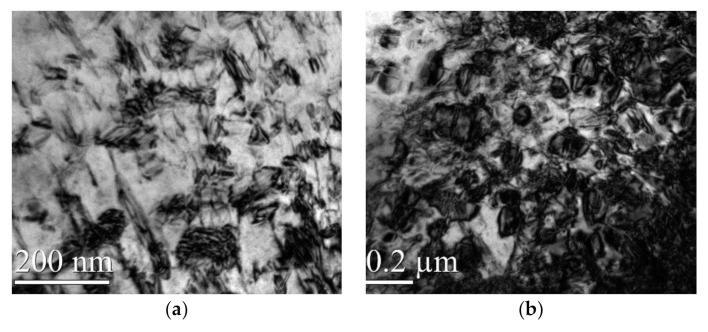
The change of the γ′′ phase shocked by LSP at different temperatures: (**a**) 25 °C, (**b**) 200 °C.

**Figure 11 materials-13-05172-f011:**
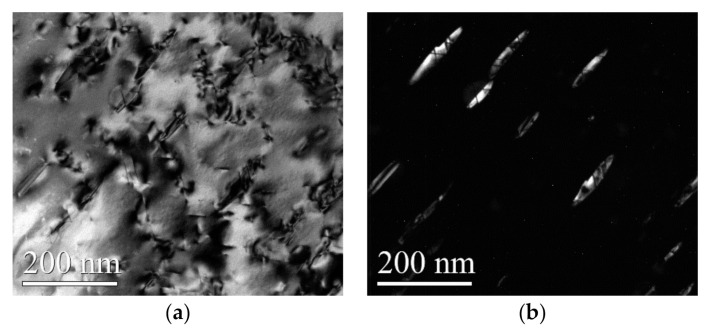
A comparison of the light field phase and the dark field phase of the δ phase shocked by LSP at 200 °C. (**a**) Bright field phase, (**b**) Dark field phase.

**Figure 12 materials-13-05172-f012:**
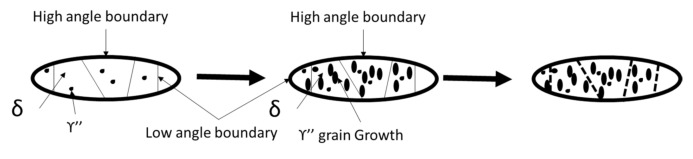
The γ″ grain deposition and the δ phase refinement in the GH4169 alloy.

**Table 1 materials-13-05172-t001:** The chemical composition of the material.

	Ni	Cr	Fe	Mo	Nb	Ti	Si
Composition (wt %)	51.58	18.36	20.17	2.96	5.03	0.99	0.18

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
