# Peer review of "Impact on Mechanical Properties and Microstructural Response of Nickel-Based Superalloy GH4169 Subjected to Warm Laser Shock Peening"

_materials, 2020, doi:10.3390/ma13225172_

Round 1

Reviewer 1 Report

Article is interesting, shows advantages of using WLSP technology for modifying surface properties. Important is that research performed by authors are related to special nickel based alloy with high application potential, especially in aviation industry.

The reviewer believes that elevating temperature of material surface not only affect on metallographic processes (alloy solution) but also on LSP efficiency (by increasing laser radiation absorptivity).

Article is suitable and recommended for publication with the following changes:

  1. Chemical composition of material should be presented in form of table.
  2. On figure 1 dimension should be added, scanning area, overlap etc.
  3. On figures 2 there is not clear that pictures shown similar area, added unevenness profile is recommended
  4. Providing explanation of using aluminum alloy as an ablative layers are necessary.
  5. On figure 5, there wis caption b) used twice , need to be corrected
  6. Incorrect figure 5 caption affect in typo in text where "...WLSP as shown in Fig. 3b (peened4 times)..." should be written "...WLSP as shown in Fig. 3c (peened4 times)..."
  7. What is the justification for the interchangeably use of energy and temperature distribution, those are two separate phenomenon?
  8. The sentence "...This nonlinear relationship between micro-hardness and plastic-deformation depth correction with the temperature and laser-induced shock pressure..." is not clear, is it written correct?
  9. Figure 4. when writing manuscript in English avoid words written in other language than English (description for measurement lines)
  10. Figure 5. it will be more clear when all three charts have same residual stress axies range for example from -900 to 0, it is hard to compare clearly differences between results when authors use different axies range.
  11. Figures 6a there is lack of scale bar.
  12. Authors specified WLSP temperature for experimental procedure 25 and 250, 300oC, if authors want to obtained some correlation between temperature during LSP process and microstructure/properties higher range of preliminary temperature might be needed, if there is literature justification of use this experimental range please provided it.

Author Response

Point 1: Chemical composition of material should be presented in form of table.

The table of chemical composition of material has been added.

Point 2: On figure 1 dimension should be added, scanning area, overlap etc.

The overlap has been added.

On figures 2 there is not clear that pictures shown similar area, added unevenness profile is recommended

Point 3: The author has compared the section drawing, from the side can not see the obvious difference, does not suggest puts in the article.

Providing explanation of using aluminum alloy as an ablative layers are necessary.

Point 4: On figure 5, there wis caption b) used twice , need to be corrected

Incorrect figure 5 caption affect in typo in text where "...WLSP as shown in Fig. 3b (peened4 times)..." should be written "...WLSP as shown in Fig. 3c (peened4 times)..."

What is the justification for the interchangeably use of energy and temperature distribution, those are two separate phenomenon?

This part has been modified the article

Point 5: The sentence "...This nonlinear relationship between micro-hardness and plastic-deformation depth correction with the temperature and laser-induced shock pressure..." is not clear, is it written correct?

This sentence has been modified. It has changed into “This relationship between micro-hardness and the temperature were nonlinear”.

Point 6: Figure 4. when writing manuscript in English avoid words written in other language than English (description for measurement lines)

This picture has been modified.

Point 7: Figure 5. it will be more clear when all three charts have same residual stress axies range for example from -900 to 0, it is hard to compare clearly differences between results when authors use different axies range.

This picture has been modified.

Point 8:Figures 6a there is lack of scale bar.

The scale bar of figures 6a was too small to see, and has been changed.

Point 9:Authors specified WLSP temperature for experimental procedure 25 and 250, 300oC, if authors want to obtained some correlation between temperature during LSP process and microstructure/properties higher range of preliminary temperature might be needed, if there is literature justification of use this experimental range please provided it.

Warm laser shock peening (WLSP) combines the advantages of both laser shock peening (LSP) and dynamic strain aging (DSA), and according to the literature, the temperature range of dynamic strain aging for nickel alloys is about.

Reviewer 2 Report

Laser shock peening (LSP) is a technique similar to shot peening that improves fatigue resistance by creating local plastic deformation, which results in the development of favorable compressive residual stresses in materials. As the authors emphasized, warm laser shock peening (WLSP) provides the advantages of both laser shock peening (LSP) and dynamic strain aging (DSA), improving the fatigue properties of materials subjected to variable loads and high temperatures.

This paper studied the surface morphology, stress and microstructure of GH4169 alloy before and after LP and WLSP processing, and also the effect of temperature on the stability of high-temperature microstructure and properties of GH4169 alloy.

The next technique and equipment were employed:

- To study the microstructure it was used Image pro plus 6.0 OM.

- Surface residual stress was measured using D/Max-2500PC X-Ray diffraction equipment.

- The microhardness of the shocked surface and cross-sectional depth was tested using a Micro Vickers Hardness Tester (DHV-1000).

- The evolution of the microstructure of the samples was investigated by TEM (JEM-2100 JEOL).

- Fracture morphology was observed using SEM (JSM-6010LA, JEOL.

- The surface morphology was observed with light interference instrument (the brand of equipment must be mentioned).

The contribution of the paper to the existing knowledge is not negligible, but the authors repeated some previously published information. For this reason, I ask for a major revision.

  1. The paper is written in an ambiguous mode in the first part, from the beginning of the Abstract to the end of the Introduction section and should be rechecked for inconsistency and other mistakes, in the present form being difficult to read.
  2. Two previously published paper by the same authors were omitted, see [1] and [2] from the Supplementary references section of the current review. The paper Hu et al. [2018] is richer in content as the actual paper, results on LSP and WLSP being presented for 25 0C, 200 0C, 250 0C, and 300 0C. From here, trying to present old results as new ones, in Introduction is stated that the paper presents work on LP processing for 25 0C, 200 0C, 250 0C only, but in Figure 4 we see results for 25 0C (written in Chinese language on the legend), 250 0C, and 300 0C. This Figure is the same as Figure 2 from Hu et al. [2018], obtained by erasing the data series for 200 0C. I kindly ask the authors to cite references [1] and [2] and to mention in the legend of Figure 2 that the information comes from [Hu et al., 2018].
  3. The Introduction section must mention the novelty of this paper in relation to the previously published ones, especially [1] and [2] from Supplementary references.
  4. All the findings and comments of the current work seem to be the original contribution of the authors, no proofs being provided for their findings and suppositions. From place to place, the comments should be supported by a citation of similar findings from the literature. For example, the first phrase from page no. 8 is “We deduced that such highly ‘‘tangled” structures would be beneficial for the stability of mechanical properties at high temperature (about 650℃).” How it was deduced such a conclusion without any evidence?
  5. Another suggestive example of such subjective interpretation can be found on page 5: “the author prefers that the theory of strengthening phase play a major role in the evolution of surface topography of GH4169”. The science is not at the free choice of the authors, this is not an acceptable scientific explanation and it should be argued more with scientific arguments, with citations, and scientific proofs.
  6. Figure 5: keep the same colors of the LSP and WLSP curves all over the subplots a, c, and c (in Figure 5c, LSP is black, and in Figures 5a and 5b, LSP is red).
  7. Also, keep the same notations all over the paper: LSP or LP, and WLP or WLSP.
  8. Figure 3: emphasize the relives formed on LSP processed surfaces.
  9. The last phrase from page 5 is incomplete.
  10. The captions of Figures 7 to 11 should specify for each sub-figure if it refers to LSP or to WLSP.

Supplementary references

[1] Hu, T., Li, S., Qiao, H., Lu, Y., Sun, B., & Wu, J. (2018). Effect of Warm Laser Shock Peening on Microstructure and Properties of GH4169 Superalloy. IOP Conference Series: Materials Science and Engineering, 423, 012054. doi:10.1088/1757-899x/423/1/012054

[2] Lu, Y., Zhao, J., Qiao, H., Hu, T., Sun, B., & Wu, J. (2019). A study on the surface morphology evolution of the GH4619 using warm laser shock peening. AIP Advances, 9(8), 085030. doi:10.1063/1.5082755

Author Response

Response to Reviewer 2 Comments

Point 1: The paper is written in an ambiguous mode in the first part, from the beginning of the Abstract to the end of the Introduction section and should be rechecked for inconsistency and other mistakes, in the present form being difficult to read.

The author makes a comprehensive revision to the article and revises the questions raised by the experts

Point 2: Two previously published paper by the same authors were omitted, see [1] and [2] from the Supplementary references section of the current review. The paper Hu et al. [2018] is richer in content as the actual paper, results on LSP and WLSP being presented for 25℃, 200℃, 250℃, and 300℃. From here, trying to present old results as new ones, in Introduction is stated that the paper presents work on LP processing for 25 â„ƒ, 200℃, 250 â„ƒ only, but in Figure 4 we see results for 25℃ (written in Chinese language on the legend), 250℃, and 300℃. This Figure is the same as Figure 2 from Hu et al. [2018], obtained by erasing the data series for 200℃. I kindly ask the authors to cite references [1] and [2] and to mention in the legend of Figure 2 that the information comes from [Hu et al., 2018].

In this paper, based on the original study, WLSP on the impact of micro-structure morphology of the law of in-depth performance, and the original work has a certain relevance, but the content of all updates, including photos are completely different choices.

Point 3: The Introduction section must mention the novelty of this paper in relation to the previously published ones, especially [1] and [2] from Supplementary references.

The supplementary references has been added.

Point 4: All the findings and comments of the current work seem to be the original contribution of the authors, no proofs being provided for their findings and suppositions. From place to place, the comments should be supported by a citation of similar findings from the literature. For example, the first phrase from page no. 8 is “We deduced that such highly ‘‘tangled” structures would be beneficial for the stability of mechanical properties at high temperature (about 650℃).” How it was deduced such a conclusion without any evidence?

The author has added references to some of the relevant literature.

Point 5: Another suggestive example of such subjective interpretation can be found on page 5: “the author prefers that the theory of strengthening phase play a major role in the evolution of surface topography of GH4169”. The science is not at the free choice of the authors, this is not an acceptable scientific explanation and it should be argued more with scientific arguments, with citations, and scientific proofs.

The author has added references to some of the relevant literature.

Point 6: Figure 5: keep the same colors of the LSP and WLSP curves all over the subplots a, c, and c (in Figure 5c, LSP is black, and in Figures 5a and 5b, LSP is red).

This Figure has been modified.

Point 7: Also, keep the same notations all over the paper: LSP or LP, and WLP or WLSP.

All abbreviations have been changed uniformly.

Point 8: Figure 3: emphasize the relives formed on LSP processed surfaces.

The relives formed surfaces by LSP has been emphasized on Figure 3.

Point 9: The last phrase from page 5 is incomplete.

This phrase has been completed.

Point 10: The captions of Figures 7 to 11 should specify for each sub-figure if it refers to LSP or to WLSP.

Figures 7 to 11 were all about WLSP, the author has revised.

Reviewer 3 Report

The authors made an investigation on the microstructure of GH4169 nickel super-alloy processed by WLSP technology with various laser parameters.

Findings:

  • Observation on proliferation and tangling of dislocations in GH4169 were carried out.
  • Dislocation density was noticed to be increased after WLSP treatment.
  • Influences of treatment (LSP and WLSP) on the microhardness distribution and along cross-sectional depth were studied.
  • TEM used to study about he microstructure evolution of GH4169 alloy produced from WLSP method.
  • Temperature effect on the stability of high temperature microstructure and properties of GH4169 alloy was also examined.

Modifications Required:

  1. The paper tends to have more extra spaces in between lines and grammatical errors.
  2. For example: (in abstract) sur-face: change to "surface"
  3. in abstract: WLP abbreviation error
  4. section 2.2: "de decided"-grammatical error.
  5. Fig 2 is not clear (can not see the numbers or quality lacks)
  6. Fig.4 please change chinese letters to English (insert letters)

The paper can be accepted after the modification.

Thank You.

Author Response

Point 1: The paper tends to have more extra spaces in between lines and grammatical errors.

For example: (in abstract) sur-face: change to "surface"

This part has been modified.

Point 2: in abstract: WLP abbreviation error

WLSP in abstract has been modified.

Point 3: section 2.2: "de decided"-grammatical error.

This grammatical error has been modified.

Point 4: Fig 2 is not clear (can not see the numbers or quality lacks)

This picture has been changed.

Point 4:Fig.4 please change chinese letters to English (insert letters)

The chinese letters in fig.4 has been changed.

Round 2

Reviewer 2 Report

The authors amended the paper as requested.

Author Response

1. Figs. 4 and 5  must be improved by adding some statistical analysis of the data: it is not required a full ANOVA, but at least a box plot, or at least some information about their variance must be added. Also, the curves are arbitrarily smoothed without any explanation: why and how this smoothing has been made? Finally, in fig.5 c the y axis is not scaled.
The Figs.4 has changed, and the curve in the Figs. 4  is not full ANOVA.
2. The microstructural analysis pèrformed via TEM is very accurate but the effect of dislocations and of the modified gamma and delta phases  are concentrated in the last three lines (371-373) with one reference: this last part is crucial for the validation of the importance of the entire paper and should be enlarged and better commented.
The author has added a further explanation here(371-373) , and was marked in red.